inorganic chemistry/organometallic chemistry/ green chemistry

deoxydehydration, catalysis, biomass upconversion, early metal-oxo

**Author for correspondence:**
Stefan M. Kilyanek
e-mail: kilyanek@uark.edu

This article has been edited by the Royal Society of Chemistry, including the commissioning, peer review process and editorial aspects up to the point of acceptance.

# Deoxydehydration of vicinal diols by homogeneous catalysts: a mechanistic overview

## Kayla A. DeNike and Stefan M. Kilyanek

Department of Chemistry and Biochemistry, University of Arkansas, 1 University of Arkansas, Fayetteville, AR 727001, USA

(iD) SMK, 0000-0002-6179-2510

Deoxydehydration (DODH) is an important reaction for the upconversion of biomass-derived polyols to commodity chemicals such as alkenes and dienes. DODH can be performed by a variety of early metal-oxo catalysts incorporating Re, Mo and V. The varying reduction methods used in the DODH catalytic cycle impact the product distribution, reaction mechanism and the overall yield of the reaction. This review surveys the reduction methods commonly used in homogeneous DODH catalyst systems and their impacts on yield and reaction conditions.

## 1. Introduction

Due to the long-term implications for the climate of continuing to consume non-renewable carbon resources, such as oil, the decarbonization of the economy has become the topic of significant public and scientific concern [1,2]. As a result, the development of processes to produce valuable carbon-based commodity chemicals from renewable carbon feedstocks has become an important field of study [3]. Currently, high-temperature steam reforming and cracking produces alkene and diene commodity chemicals from petroleum [4]. Fortunately, renewable biomass-derived materials may serve as a sustainable alternative to produce these carbon-based commodity chemicals currently produced from petrochemical refinement [5,6]. One significant challenge for upconverting biomass into commodity chemicals is the relatively large number of oxygen functionalities present in biomass-derived material [7]. Development of reactions to combat this challenge and upconvert biomass-derived material has been of great interest in recent years [8].

One route to biomass upconversion is the generation of polyols from cellulose. Cellulose may undergo hydrolysis to produce polyols and carbohydrates such as sorbitol and

**Scheme 1.** Hydrolysis of cellulose to form sorbitol.

Z = sacrificial reductant

**Scheme 2.** Deoxydehydration converts vicinal diols and a sacrificial reductant to alkene, water and oxidized reductant.

**Scheme 3.** Generic reaction mechanism for DODH.

mannitol (scheme 1) [9]. Additionally, products such as erythritol can be produced by the decarbonylation of pentoses or by the fermentation of glucose [10,11].

The search for an efficient way to convert renewable carbohydrate feedstocks to materials and fuels has inspired research towards developing catalysts that can dehydrate these oxygen-rich substrates. Here, we will focus specifically on the development of and advances in the deoxydehydration (DODH) of polyols to generate alkenes and dienes (for other reviews of DODH, see [12–20]).

The generic reaction scheme for DODH is shown in scheme 2. DODH is typically catalysed by a metal-oxo catalyst coupled with a sacrificial reductant. DODH converts vicinal diol functionalities into alkenes while also producing water and consuming/oxidizing a sacrificial reductant.

DODH can be thought of as the reverse reaction of the dihydroxylation of alkenes by metal oxides such as $OsO_4$ [21,22]. Alternatively, DODH can be viewed as an overall dehydration combined with a net oxygen-atom abstraction. DODH is of particular interest because a single olefinic product can be obtained by combining dehydration and deoxygenation into a single catalytic cycle. By contrast, the dehydration of polyols often produces complex product mixtures [23].

A general mechanism for the DODH of vicinal diols by metal-dioxo fragments is summarized in scheme 3. DODH begins with the condensation of the diol with a metal-oxo bond, releasing water. The metal-oxo-diolate is then reduced via a sacrificial reductant, resulting in the formation of a reduced metal diolate, and the oxidized reductant. Finally, olefin extrusion, the likely rate-limiting step [24], produces alkene and regenerates the metal-dioxo catalyst.

Chart 1 provides a non-exhaustive survey of metal-oxo catalysts used in homogeneous DODH reactions. Two common features of these catalysts are: pre-catalysts are $d^0$ metal centres and pre-catalysts contain cis-dioxo-metal moieties. As shown in chart 1, a limited number of ligand environments have been surveyed to date.

In this review, we will discuss the differing mechanistic pathways for homogeneously catalysed deoxydehydration and how these differences influence reactivity and product distribution. We have generally categorized the various mechanistic pathways by the mechanism of reduction: (i) oxo-abstraction by phosphines, sulfites and elemental reductants, (ii) transfer hydrogenation from

**Re**

R = Me, Cp*, Cp$^{R3}$

$Re_2O_7$

$[G][ReO_4]$

G = $NH_4^+$, $Na^+$, $NBu_4^+$

$[trans\text{-}ReO_2(py)_4][X]$

X = $Cl^-$, $PF_6^-$

**Mo**

$[NH_4]_6[Mo_7O_{24}]$

R = Me, t-Bu

R = Me, Cl
L⌒L = bipy, DME

R = $C_4H_4S$

Fristrup [72]

Stalpaert & De Vos [46], Hills et al. [44]

Fristrup [72]

Hills et al. [44]

**V**

G = $NBu_4^+$

G = $NBu_4^+$
R = $C_4H_4S$

$[G][VO_3]$

G = $NH_4^+$, $Na^+$

Nicholas [47,48]

Fristrup and co-workers [78]

**Chart 1.** Current DODH catalysts.

sacrificial alcohols and other substrates, and (iii) reduction of metal by the oxidative cleavage of diol substrates. Additionally, we have grouped DODH catalyst systems by transition metal within each category. Finally, we will discuss cases where multiple mechanistic pathways are apparently active.

# 2. Reduction mechanism: oxo-abstraction

The first reports of DODH in the literature exploited oxygen-atom abstraction to generate a reduced metal species. Oxo-abstraction by a variety of reductants, including phosphines, sulfites and elemental reductants such as Zn, are all found in the literature. DODH systems exploiting oxo-abstraction have been found to be robust and show good catalytic performance.

## 2.1. Rhenium catalysts

### 2.1.1. Phosphines as reductants

Among the most studied DODH catalyst systems are Re catalysts coupled with phosphine reductants. DODH was first reported by Cook & Andrews [24]. Using Cp*ReO$_3$ and aryl phosphines as reductants, they demonstrated catalytic activity at elevated temperatures and produced alkenes in good yield. They proposed a mechanism consistent with initial oxo-abstraction and reduction of the Re(VII) trioxo catalyst by a phosphine followed by condensation of the diol with an M=O bond of the Re(V) species (scheme 4, right-hand path). The resulting metal diolate is then thought to undergo olefin extrusion to release olefin and regenerate the Re(VII) catalyst. Additionally, Cook and Andrews postulated that further reduction to Re(III) leads to catalyst deactivation. The order of the individual steps in the reaction is a topic of debate; reduction can occur either before or after diolate condensation, and the order may be affected by the identity of the reductant or substrate [25].

Scheme 4 summarizes the two possible reaction pathways discussed above. Each pathway contains an Re(VII)/Re(V) redox cycle. Olefin extrusion is proposed to be the rate-limiting step in both

**Scheme 4.** Possible DODH reaction mechanisms of RReO$_3$ exploiting reduction by oxo-abstraction.

**Scheme 5.** Proposed mechanism for olefin extrusion.

pathways [26]. Gable & Juliette studied olefin extrusion from rhenium diolates extensively. Re diolates analogous to those proposed to exist in the catalytic cycle of DODH were found to undergo cycloreversion through methylene migration to form an Re(VII) metallaoxetane intermediate (scheme 5). This cycloreversion to form metallaoxetane appears to occur instead of a concerted cycloreversion step to directly release alkene. These studies implied that diolate substituents that form staggered conformations form olefins at faster rates. These findings explain the apparent preference for forming internal E-alkenes via DODH.

The proposed reaction steps for DODH were not unprecedented. Herrmann *et al.* [27] had previously demonstrated oxo-abstraction of Cp*ReO$_3$ by phosphines to form the oxo-bridged dimeric rhenium species (Cp*)$_2$Re$_2$O$_4$. Condensation between methyltrioxorheynium (MTO) and catechols had also been previously demonstrated [28]. Finally, Gable [29] had independently synthesized reduced Cp*ReO(diolate) species and found that upon heating, the diolate undergoes cycloreversion to form alkene and Cp*ReO$_3$, successfully demonstrating the final step of the DODH reaction several years before the report of the catalytic reaction. Phosphines continue to be investigated as reductants in DODH reactions catalysed by Re compounds. Phosphine-driven Re-catalysed DODH remains an active area of research with an ever-expanded substrate scope that includes 1,2-octanediol [30–32], 1,2,5,6-diisopropylidene-D-mannitol [24], glycerol [24], 1,2-butanediol [24] and 1-phenyl-1,2-ethanediol [24].

### 2.1.2. Sulfites as reductants

Although phosphines were found to be very effective at DODH catalyst turnover, other, less costly reductants were of interest. Molybdenum enzymes were known to be oxidized by simple sulfites, suggesting that sulfites might be used as an oxo-acceptor to promote DODH reactions [33]. Indeed, sulfite reduction of Re-oxo catalysts was shown by Nicholas and co-workers to promote DODH catalyst turnover. Interestingly, sulfites were found to prefer to react with *syn* diols and form alkenes; *anti* diols were found to give poor yields [34,35]. The DODH of a wide variety of diols driven by sulfite oxidation has been demonstrated using several rhenium catalysts with yields up to 89%. Computational studies suggest that sulfite-driven DODH proceeds first by oxo-abstraction followed by condensation to the diolate and olefin extrusion [36]. This mechanistic study also suggested that when MTO is the catalyst,

weakly coordinating oxidized sulfate by-products offer a more active catalytic system than strongly coordinating reductants such as phosphines. It was proposed that strong coordination of reductants and their by-products to the metal centre raises the activation energy for olefin extrusion [36].

### 2.1.3. Other oxo-abstraction reagents

Elemental reductants such as metallic Zn and Fe have been found to be competent oxo-abstracting agents. McClain & Nicholas showed zinc, carbon, iron and manganese act as elemental reductants for the DODH of polyols using ammonium perrhenate (APR) affording upwards of 90% yield for a variety of substrates [37]. These reductants were screened with simple Re compounds, including the chloride and hexafluorophosphate salts of *trans*-[ReO$_2$(Py)$_4$]$^+$, which produced 1-decene from 1,2-decanediol in 90% and 67% yields, respectively.

To date, catalysts with relatively simple structures have been used to afford DODH. For example, APR and other simple rhenium compounds such as Re$_2$O$_7$ [30,38], NaReO$_4$ and [NBu$_4$][ReO$_4$] have been shown to catalyse DODH using a variety of reductants and a plethora of diols [39,40]. Finally, MTO and Cp*ReO$_3$ remain the most widely used DODH catalysts today often coupled with oxo-abstracting reagents. A growing number of substrates, including tartaric acid [40], 1,2-octanediol [34,35], 1-phenyl-1,2-ethanediol [34,35], 1,2-cyclohexanediol [34], 1,2-decanediol [35] and 1,2-tetradecanediol [35], can be converted to alkenes in good to excellent yields.

## 2.2. Molybdenum

Mo(VI)/Mo(IV) catalytic cycles incorporating the breaking and making of Mo=O bonds have been found in a number of metalloenzymes [41,42]. Oxo-abstraction of Mo-oxo species by phosphines has been shown to proceed via a nucleophilic attack and disassociation mechanism where the association of the phosphine to the metal-oxo is the rate-limiting step. This step was found to be sensitive to the sterics of the phosphine and has a linear correlation to phosphine cone angle [43]. These known catalytic cycles and reactions make Mo an enticing platform for DODH.

Chart 1 (vide supra) includes a survey of Mo systems shown to perform DODH. All the catalysts shown can achieve catalysis using a variety of reduction pathways including oxo-abstraction. The first reported DODH of vicinal diols using a molybdenum catalyst was published by Hills *et al.* [44]. Acylpyrazolonate-dioxomolybdenum(VI) Mo(O)$_2$(QR)$_2$ complexes (chart 1) were found to deoxygenate epoxides catalytically using PPh$_3$ as reductant. Additionally, these complexes performed DODH of 1-phenyl-1,2-ethanediol and cyclooctanediol in modest yields (13%, 55%) in toluene at 110°C using PPh$_3$ as oxo-acceptor.

Navarro & John [45] have demonstrated that commercially available ammonium heptamolybdate (AHM) is a competent DODH catalyst using a variety of reductants including Na$_2$SO$_3$ and PPh$_3$. AHM is a competent catalyst in toluene at 170–190°C and affords modest alkene yields (23%) for most substrates.

The influence of supporting ligands on molybdenum-catalysed DODH has been of recent interest [46]. 2,2,6,6-Tetramethylheptanedionate (TMHDH) was used to generate a complex mixture of active catalysts *in situ*. When using a more sterically demanding catalyst environment compared to MoO$_2$(acac)$_2$, the TMHDH complex was found to afford a higher alkene yield of 93% for 1-hexene produced from 1,2-hexanediol using PPh$_3$ as reductant. It is postulated that steric and electronic ligand effects influence the product distribution produced by these catalyst mixtures. Electron-withdrawing ligands, such as 1,1,1,5,5,5-hexaflouroacetylacetonate and 1,1,1-triflouroacetylacetonate, decrease the basicity of the ligands and thus appear to quench the oxophilicity of molybdenum and hinder olefin extrusion. Stalpaert & De Vos [46] suggest electron-donating ligands help lower the barrier to olefin extrusion; they state that by increasing the electron density at the metal centre, they expect the electron transfer from Mo to diolate to be faster overall. Finally, they propose that Mo catalysts with sterically bulky ligands such as TMHDH and dibenzoylmethane should disfavour dimerization and oligomerizion under catalytic conditions, thereby preventing catalyst deactivation. These *in situ*-generated catalysts were also found to be competent catalysts using a variety of other oxo-acceptor reagents including Na$_2$SO$_3$ and CO.

## 2.3. Vanadium

In 2013, Chapman & Nicholas [47] demonstrated the first example of vanadium-catalysed DODH using oxo-abstracting reductants. 1-Phenyl-1,2-ethanediol was converted to styrene in 95% yield using

**Scheme 6.** DODH applied in a multi-step organic synthesis sequence.

tetrabutylammonium dioxovanadium(V)-dipicolinate, ([NBu$_4$][VO$_2$(dipic)]), with PPh$_3$ as reductant. Furthermore, Na$_2$SO$_3$ was found to be a superior reductant, affording shorter reaction times with comparable (87%) yield. The proposed mechanism is analogous to that for the previously studied Re systems shown in scheme 3 but incorporates a V(V)/V(III) redox cycle. Additionally, Gopaladasu & Nicholas [48] demonstrated that CO was a competent oxo-abstraction reagent that affords alkene yields of up to 97%.

Debate over which pathway the reaction follows has led to investigations of the mechanism using density functional theory (DFT) [49]. DFT calculations suggest that the vanadium-dioxo catalyst undergoes oxo-abstraction followed by condensation of the diol forming the reduced metal diolate species [49]. The barrier to diol condensation with V(V)-dioxo species was found to be higher in energy (39.8 kcal mol$^{-1}$) than condensation with a V(III)-mono-oxo centre (13.4 kcal mol$^{-1}$). DFT studies proposed a high spin V(III)-diolate species as the critical intermediate that affords olefin extrusion.

Geary *et al.* [50] used Nicholas' V-catalysed sulfite-driven DODH in a multi-step synthetic scheme. Ruthenium-catalysed diol–diene [4 + 2] cycloaddition was combined with vanadium-catalysed DODH and aerobic dehydrogenative aromatization to yield acenes (scheme 6). [NBu$_4$][VO$_2$(dipic)] with Na$_2$SO$_3$ was also found to transform tetraols into tetraenes at up to 87% yield by sequential DODH at 180°C. These applications demonstrate that V-catalysed DODH can be used with a variety of substrates and can generate internal alkenes as well as the terminal alkenes typically formed in DODH studies.

## 2.4. Summary of DODH reactions using oxo-abstraction reagents as reductants

Table 1 summarizes the catalyst systems with the highest yields using oxo-abstraction reagents. These results show that oxo-abstraction reagents are effective in driving catalyst turnover for DODH and afford high olefin yields at relatively mild reaction temperatures.

# 3. Reduction mechanism: transfer hydrogenation

Transfer hydrogenation mechanisms for metal-oxo reduction have been used in DODH reactions with excellent results. Alcohols have been found to successfully reduce metal-oxo catalysts for DODH by an effective two-proton two-electron reduction of the metal-oxo-bond to generate water. Typically, secondary alcohols are used as sacrificial reductants in DODH reactions giving ketones as a reaction by-product. The advantages of using secondary alcohols as reductants are that they are cheaper and more easily recycled than reductants such as phosphines and sulfites [51]. A generic scheme for metal-oxo reduction by transfer hydrogenation from a secondary alcohol is shown in scheme 7.

## 3.1. Rhenium

Arceo *et al.* [52] investigated the didehydroxylation of vicinal diols to alkenes using Re$_2$(CO)$_{10}$ and a secondary alcohol as a reductant. This was the first reported DODH of diols using a secondary sacrificial alcohol as a reductant. They initially found that without a secondary alcohol as a sacrificial reductant, the diol substrate was converted to an alkene and the corresponding diketone product formed from transfer hydrogenation from the diol. To prevent consumption of substrate, a secondary alcohol was added as a sacrificial reductant. Re$_2$(CO)$_{10}$ was found to be an effective catalyst for DODH of 1,2-tetradecanediol using a variety of reductants, including 5-nonanol, 3-octanol and 2-octanol. Alkene was obtained in up to 84% yield. With the addition of *para*-toluenesulfonic acid (TsOH), conversion reached 100%. Initial screening of the biomass-derived substrate erythritol showed cyclization and deoxydehydration to 2,5-dihydrofuran in 62% yield.

Shiramizu & Toste [53] postulated that the active rhenium species during Ellman and Bergman's DODH reactions is an oxidized species, since Re$_2$(CO)$_{10}$ only converts diol to olefin in the presence of

**Table 1.** Summary of DODH reactions using oxo-abstraction reagents.

| catalyst | reductant | temp (°C) | substrate | product | yield (olefin) (%) | ref. |
|---|---|---|---|---|---|---|
| MTO | PPh$_3$ | 165 | 1,2,6-HT | 4-penten-1-ol | 96 | [38] |
| Cp$^{tt}$ReO$_3$ | PPh$_3$ | 135 | 1,2-octanediol | 1-octene | 93 | [32] |
| Re$_2$O$_7$ | PPh$_3$ | 165 | 1,2,6-HT | 4-penten-1-ol | 77 | [38] |
| [n-Bu$_4$N][ReO$_4$] | P(o-tolyl)$_3$ | 150 | 1-phenyl-1,2-ethanediol | styrene | 70 | [35] |
| [n-Bu$_4$N][ReO$_4$] | Na$_2$SO$_3$ | 150 | 1,2-decanediol | 1-decene | 89 | [35] |
| [(Py)$_4$ReO$_2$]Cl | Zn | 150 | 1,2-decanediol | 1-decene | 90 | [37] |
| MoO$_2$(acac)$_2$ + TMDH | PPh$_3$ | 200 | 1,2-hexanediol | 1-hexene | 93 | [46] |
| MoO$_2$(Qhe)$_2$ | PPh$_3$ | 110 | 1-phenyl-1,2-ethanediol | styrene | 86 | [44] |
| [NBu$_4$][(dipic)VO$_2$] | PPh$_3$ | 170 | 1-phenyl-1,2-ethanediol | styrene | 95 | [47] |
| [NBu$_4$][(dipic)VO$_2$] | PPh$_3$ | 170 | 1,2-octanediol | 1-octene | 97 | [47] |
| [NBu$_4$][(dipic)VO$_2$] | Na$_2$SO$_3$ | 170 | 1-phenyl-1,2-ethanediol | styrene | 87 | [47] |
| [NBu$_4$][VO$_2$ (salicylaldehyde hydrazide)] | CO | 180 | 1,2-hexanediol | 1-hexene | 48 | [48] |

**Scheme 7.** Generic mechanism for transfer hydrogenation from a secondary alcohol to a metal-dioxo fragment.

**Scheme 8.** MTO-catalysed DODH of $C_4$ tetraols yielding 1,3-butadiene.

air. Using MTO, Toste and co-workers found that large, biomass-derived sugar alcohols could be converted to alkenes and dienes using simple secondary alcohols as sacrificial reductants. A number of alcohols can afford catalyst turnover in DODH systems, including 1-butanol (a biomass-derived alcohol) and 3-octanol, which produced better results. MTO and 3-octanol successfully converted glycerol to allyl alcohol in 90% yield. Additionally, erythritol was converted to the industrially important 1,3-butadiene in 89% yield with 11% yield of 2,5-dihydrofuran also observed. DL-threitol also could be reduced to 1,3-butadiene in 81% yield while producing 1,4-anhydroethreitol (scheme 8).

These results imply that *syn* diols undergo DODH faster than *anti* diols since DL-threitol produced 1,4-anhydroethreitol and not 2,5-dihydrofuran. A screen of substrate scope finds that a number of biomass-derived polyols such as xylitol, D-arabinitol, ribitol, D-sorbitol, D-mannitol and a variety of inositols can undergo reduction using secondary alcohols and MTO. Other notable reactions include the conversion of *myo*-inositol to benzene and the conversion of tetroses and hexoses to furans in moderate yield [53].

Shiramizu & Toste [54] expanded on these findings and showed successful DODH of other relevant biomass-derived polyols including mucic acid, mucic acid dibutyl ester, gluconic acid, L-(+)-tartaric acid, erythritol, D-erythronolactone and D-(+)-ribono-1,4-lactone into useful commodity chemicals including plasticizer precursors.

Ison and co-workers [55] investigated the mechanism of DODH using secondary alcohols as reductants for MTO and found many bridging diolate intermediates. The rate of reaction was found to be zeroth order in substrate, and half order in Re. The active species was found to be the reduced methyldioxorhenium (MDO) species instead of MTO. MDO can undergo condensation to form an Re(V) diolate, which is reduced to an Re(III) diolate that undergoes olefin extrusion. Upon reduction to the Re(III) diolate, an oxo-bridged dimeric Re(IV) diolate species forms; this is probably the source of the half-order dependence on Re. Dimer formation leads to decreased catalytic activity because the reaction rate depends on scission of the dimer before olefin extrusion. A simplified mechanism showing only the critical species and equilibrium is shown in scheme 9.

Abu-Omar and co-workers [56] investigated the transfer hydrogenation of glycerol to afford the DODH product, allyl alcohol and other side products. MTO and [NH₄][ReO₄] were both found to catalyse the conversion of glycerol into allyl alcohol and products from transfer hydrogenation and dehydration including propanal, dihydroxyacetone and acrolein. In these reactions, glycerol is both the solvent and substrate, thereby simplifying the separation of volatile products. The total conversion of the reaction was found to be 74% with allyl alcohol as the major product. The mechanism of glycerol reduction is shown in scheme 10. Olefin extrusion is the rate-limiting step in the DODH reaction, so hydride transfer to the rhenium diolate from a second equivalent of glycerol forms MTO and either 1,3-propanediol or 1,2-propanediol, which can produce propanal and acrolein by metal-catalysed deoxygenation or dehydration [57].

Notably, when sacrificial reducing alcohols are used as solvents, glycerol still competes with the alcohol as a transfer-hydrogenation agent, forming acrolein or propanal. This was true for a number of sacrificial alcohols including: 3-octanol, 1-heptanol, 1-cyclohexanol, 1,3-propanediol or 1,2-propanediol. The use of a sacrificial alcohol solvent improved the overall yield of allyl alcohol to

**Scheme 9.** Rhenium(III)/rhenium(V) diolate dimerization equilibrium generating a half-order dependence of Re in the rate law for DODH reduced by secondary alcohols.

**Scheme 10.** Side product formation during the DODH of glycerol by MTO [56]. Primary DODH cycle is shown in blue; minor pathways to produce side products are shown in red. Propanol and acrolein were proposed to form by metal-catalysed deoxygenation/dehydration [57].

by-products; however, glycerol is still a competent reductant in the presence of added primary alcohols. Alkene formation from the sacrificial alcohol solvent also occurred and is a result of the known rhenium-catalysed dehydration of alcohols to olefins under the catalytic conditions used [58,59].

Boucher-Jacobs & Nicholas [60] used a primary alcohol as reductant to allow for easier separation of the corresponding aldehydes produced by transfer hydrogenation. Benzyl alcohol and APR were found to catalyse the DODH of a variety of polyols at 140–175°C in excellent yield (up to 95%). Convenient separation of catalyst was achieved by filtration followed by precipitation of benzaldehyde with bisulfate.

Abu-Omar and co-workers [61] found that reaction of MTO and $H_2$ catalysed the deoxygenation of epoxides and the DODH of diols. Using MTO, a number of diols could be deoxygenated to alkenes, which are then hydrogenated to alkanes under catalytic conditions (80–300 psi at 150°C) in some cases. This diol-to-alkene reaction was applied to the biomass-derived diol anhydroerythritol to afford dihydrofuran in 25% yield. Abu-Omar *et al.* propose a mechanism in which condensation of the diol onto the reduced MDO forms the rhenium mono-oxo-diolate and is followed by metallaoxetane formation and olefin extrusion in a fashion analogous to the process in scheme 5.

Rhenium-catalysed DODH using alcohols as reductants remains a topic of significant interest and to date has been demonstrated with a variety of substrates not described in detail here [62–65]. Finally, hydroaromatics and indolines have been demonstrated to be competent transfer hydrogenation agents that can drive DODH with MTO in excellent yields (greater than 90%) for a variety of substrates [66,67].

## 3.2. Molybdenum

Transfer hydrogenation catalysed by molybdenum complexes has been studied for decades [68–70]. With this precedent, Beckerle *et al.* [71] used Mo-dioxo-bis(phenolate) ligands to catalyse the DODH of diols using 3-octanol as the reductant. Anhydroerythritol was converted to 2,5-dihydrofuran in 37–57% yield. In the presence of acid, a variety of side products are produced, including 3-octene produced by the dehydration of 3-octanol at high temperatures. However, the reaction temperature can be decreased when microwave radiation is used to enhance the reaction rate.

Fristrup and co-workers [72] demonstrated that isopropyl alcohol could be used as a sacrificial reductant to perform DODH of 1,2-decanediol. The reduction of diols generally formed alkenes in approximately 50% yield at 5 mol% catalyst. Alkene yields were found to rise to 70–77% when bases such as $NBu_4OH$ were added to the reaction mixture. In this study, a variety of diols were screened for DODH activity and found to have similar alkene yields. Cyclic diols, however, were found to have appreciably lower yields (approx. 20%) using these simple catalysts.

## 3.3. Vanadium

Fristrup *et al.* also investigated the DODH of glycerol as both substrate and reductant using simple vanadium salts, such as ammonium vanadate ($NH_4VO_3$) as catalysts. Allyl alcohol was found in modest yields of approximately 20% and a variety of side products are also formed by alcohol dehydration reactions occurring under catalytic conditions. Gopaladasu & Nicholas [48] were able to achieve approximately 50% yield of alkene using benzyl alcohol with a variety of vanadium coordination compounds. Additionally, Gopaladasu & Nicholas found that hydrogen could serve as a competent reductant, producing alkenes in yields as high as approximately 40% [48].

## 3.4. Summary of DODH reactions using transfer hydrogenation

Table 2 summarizes the catalyst systems with the highest yields using secondary alcohols and other transfer hydrogenation agents. Yields of DODH reactions driven by transfer hydrogenation vary from poor to excellent. Once again, Re systems show excellent performance.

# 4. Reduction mechanism: oxidative cleavage of diols and multiple mechanisms

Although oxo-abstraction reagents are clearly effective at affording DODH catalyst turnover, these reagents provide poor atom economy [74]. Oxidative cleavage of diols has been found to occur in some catalyst systems. Oxidative cleavage of substrate occurs when a metal-oxo-diolate undergoes C–C bond cleavage to yield two equivalents of aldehyde or ketone and a reduced metal centre (scheme 11). When a diol is used as both the substrate and reductant in DODH reactions, alkenes can be afforded in a maximum yield of 50% while half of the substrate is consumed to reduce the metal centre and generate aldehydes (scheme 11).

Oxidative cleavage of diols allows the catalytic reaction to be performed in neat substrate thereby removing the need for solvent. While this has obvious advantages, it does require the separation of reactants and products, typically by distillation. Additionally, acetal can be produced from the

**Table 2.** Summary of DODH reactions using transfer hydrogenation reagents.

| catalyst | reductant | temp (°C) | substrate | product | yield (olefin) (%) | ref. |
|---|---|---|---|---|---|---|
| APR | 3-octanol | 150 | 1,2-tetradecanediol | 1-tetradecene | 99 | [65] |
| MTO | 3-octanol | 140 | (R,R)-(+)-hydrobenzoin | trans-stilbene | 80 | [55] |
| MTO | 3-pentanol | 170 | anhydroerythritol | 2,5-dihydrofuran | 95 | [53] |
| MTO | 3-pentanol | 170 | $C_6$ sugar acohols | (E)-hexatriene | 54 | [53] |
| Re₂(CO)₁₀ | 3-octanol | 170 | anhydroerythritol | 2,5-dihydrofuran | 91 | [53] |
| MTO | H₂, 1,3-propanediol | 140 | glycerol | allyl alcohol | 91 | [73] |
| AHM | iPrOH | 240–250 | 1,2-decanediol | 1-decene | 55 | [72] |
| AHM + [NBu₄][OH] | iPrOH | 240–250 | 1,2-hexanediol | 1-hexene | 77 | [72] |
| MoO₂Cl₂(bipy) | iPrOH | 240–250 | 1,2-decanediol | 1-decene | 46 | [72] |
| bis(phenolato)MoO₂ | 3-octanol | 200 | anhydroerythritol | 2,5-dihydrofuran | 49 | [71] |
| [NBu₄][VO₂ (salicylaldehyde hydrazide)] | benzyl alcohol | 180 | 1,2-hexanediol | 1-hexene | 48 | [48] |
| [NBu₄][VO₂ (salicylaldehyde hydrazide)] | H₂ | 180 | 1,2-hexanediol | 1-hexene | 33 | [48] |

**Scheme 11.** Generic scheme for DODH facilitated by oxidative deformylation of vicinal diols. Reactants are shown in blue; products in red.

**Scheme 12.** Oxidative deformylation of glycerol and acetal formation of diols [75].

condensation of diol with the aldehyde products, both reducing the overall alkene yield and further complicating the product mixture.

## 4.1. Rhenium

Oxidative cleavage of diols is not commonly observed for Re systems. This predilection makes Re catalysts ideal candidates for DODH catalysed by transfer hydrogenation from sacrificial alcohols without the generation of side products (vide supra). However, aromatic diols have been shown to undergo oxidative cleavage reactions with $ReO_2$(diolate) species. As mentioned previously, Abu-Omar and co-workers [55] found a complex mixture of diolate intermediates when studying the mechanism of MTO-catalysed DODH with secondary alcohols. When using (*R*,*R*)-(+)-hydrobenzoin as substrate, oxidative cleavage of the Re(V)-diolate species generates two equivalents of benzaldehyde and the catalytically active species, methyldioxorhenium (MDO). To our knowledge, this is the only case of oxidative cleavage of a diol in an Re-catalysed DODH cycle.

## 4.2. Molybdenum

Fristrup and co-workers [75] studied DODH reactions using molybdenum catalyst systems driven by oxidative cleavage. $(NH_4)_6Mo_7O_{24}$ (AHM) was used as a catalyst with multiple aliphatic diols including glycerol at 195–220°C. Under catalytic conditions, several acetals formed through condensation of unreacted diol with the aldehyde by-products (scheme 12 reactions i–iii). This scheme highlights that oxidative cleavage of diols results in complicated product mixtures. These product mixtures are a result of the subsequent condensation reactions possible that limit yield and complicate product separation.

When glycerol was used as substrate with AHM, allyl alcohol was produced in very low yields (less than 9% [75]). Significant yields of side product and the secondary reaction products glycoaldehyde and formaldehyde (scheme 9 reaction iv) were observed in the product mixture. The alkene yield improved

**Table 3.** Summary of DODH reactions using oxidative cleavage.

| catalyst | notes | temp (°C) | substrate | product | yield (olefin) (%) | ref. |
|---|---|---|---|---|---|---|
| MTO | — | 140 | (R,R)-(+)-hydrobenzoin | trans-stilbene | 50 | [55] |
| MoO$_2$Cl$_2$(bipy) | | 220 | 1,2-hexanediol | 1-hexene | 19 | [75] |
| AHM | 1,5-pentanediol as solvent | 220 | 1,2-hexanediol | 1-hexene | 45 | [75] |
| AHM | 1,5-pentanediol as solvent | 220 | glycerol | allyl alcohol | 40 | [75] |
| NH$_4$VO$_3$ | | 275 | glycerol | allyl alcohol | 22 | [78] |
| V$_2$O$_5$ | | 275 | glycerol | allyl alcohol | 22 | [78] |

upon using 1,5-pentanediol as substrate and reductant. Oxidative cleavage of vicinal diols has also been reported in the literature. Oxo-donors such as DMSO can be used to re-oxidize the reduced Mo centres by oxo transfer, implying that these reactions are reversible [76]. Although DODH systems that exploit oxidative cleavage cannot have the same high yields as the oxo-abstraction reactions discussed above, the by-products formed are sometimes useful commodity chemicals and can be separated by distillation, thereby producing a second value-added organic compound to the product mixture [77]. Additionally, the diol itself is the reductant because the same product distribution is found when heating AHM and 1,2-decanediol under N$_2$ or H$_2$; AHM and 1,2-decanediol in hexane at 247°C under 22 bar N$_2$ afforded decene in 55% yield (in these cases, it can be a challenge to differentiate between the oxidative cleavage of diol and transfer hydrogenation of diol with minor products) [72].

## 4.3. Vanadium

Fristrup and co-workers [78] also studied the DODH of glycerol to form allyl alcohol using simple vanadium catalysts and no external reductants. [NH$_4$][VO$_3$] can catalyse DODH of glycerol at significantly higher temperatures than previously observed with MTO (275°C and 165°C, respectively). [NH$_4$][VO$_3$] afforded a decent yield of allyl alcohol compared to MTO in the absence of external reductants (22 versus 12%) [73,79].

## 4.4. Multiple mechanisms

While characterization and description of the DODH reduction mechanisms found in the literature is illuminating and provides insights into designing new catalyst systems, some of the excellent work described here does in fact show several competing mechanisms for catalyst reduction. For example, Fristrup and co-workers always observe complicated product mixtures from both the reaction of catalytically generated products with starting materials and the generation of aldehyde products by the seemingly ubiquitous oxidative cleavage of high oxidation state Mo and V diolates. Yields of aldehyde can surpass 20% even at temperatures approaching 200°C with sacrificial alcohol as solvent [75]. Oxidative cleavage of diols even competes with reduction of MTO systems when using (R,R)-(+)-hydrobenzoin [56]. Additionally, glycerol was found to act as a reductant by transfer hydrogenation and by oxidative cleavage for V [78] and Mo [72], respectively, and both metals have precedent for performing both reactions by being compatible with both pathways. Care must be taken when analysing product mixtures to account for the possibility of several competing reduction reactions.

## 4.5. Summary of DODH reactions using the oxidative cleavage of diols as reductant

Table 3 summarizes the catalyst systems with the highest yields using oxidative cleavage to drive DODH. These results show that deformylation seems to be ubiquitous in V and Mo systems, whereas Re does not commonly show this reactivity.

# 5. Conclusion

Rhenium catalysts have been shown to be the most efficient catalysts for DODH by affording both fast reaction rates and excellent alkene yields. Although rhenium is a rare and expensive metal, the reaction conditions required for DODH using rhenium catalysts are milder than the conditions required for current molybdenum and vanadium catalyst systems. Rhenium systems are also not hampered by multiple competing reaction mechanisms; probably because of the relatively low temperatures required to perform catalysis. Although significant progress has been made exploring other metals and ligand environments, MTO is still the reigning champion in the field of DODH, affording fast catalyst turnover with nearly all reductants studied at relatively low temperatures.

Finally, it is noteworthy that there are relatively few catalyst systems in the literature. A great deal of exploration is needed in the chemical space of molybdenum- and vanadium-oxo catalysts, including manipulation of the steric and electronic environments produced by the ligands. Only a few catalysts have been explored, and, especially for molybdenum, their alkene yields and selectivities require significant improvement to reach the activity of Re, so continued work may well find superior systems. Molybdenum-oxo compounds are particularly promising because a large number of complexes can be synthesized readily from easily accessible starting materials. Vanadium's possibilities are also quite intriguing, because it can adopt a number of spin states in reduced species.

Data accessibility. This article has no additional data.

Authors' contributions. Both authors contributed equally to this work.

Competing interests. The authors declare no competing interest.

Funding. This work was supported by the National Science Foundation, award no. CHE-1654553.

Acknowledgements. S.M.K. thanks Dr Daniel C. O'Hanlon for useful conversations.

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
