## [Reviewer comments · Royal Society Open Science]

Review History

RSOS-191165.R0 (Original submission)

Review form: Reviewer 1

Is the manuscript scientifically sound in its present form?

Yes

Are the interpretations and conclusions justified by the results?

Yes

Is the language acceptable?

Yes

Do you have any ethical concerns with this paper?

No

Have you any concerns about statistical analyses in this paper?

No

Recommendation?

Accept with minor revision (please list in comments)

Comments to the Author(s)

Kilyanek and coworkers present in their manuscript "deoxydehydration of vicinal diols by homogeneous catalysis: A mechanistic overview" a relatively narrow chapter of deoxydehydration of polyols to generate alkenes and dienes chemistry, but in a very comprehensive. The review is quite well written and the collection of literature is very well presented. The authors described the mechanistic pathways for these DODH reactions. The description and representation of the examples in the manuscript does not need to be criticized. It is rather the selection.

Given the topical interest, with the practical importance of sustainable deoxydehydration reactions, I recommend publication of this fine manuscript after the following revision has been corrected.

1) Scheme 9, the drawing could be corrected.

2) Reference 42 and 61 are same: this need to be corrected.

There are several minor formatting errors throughout the manuscript which could be corrected by careful editing.

Review form: Reviewer 2

Is the manuscript scientifically sound in its present form?

Yes

Are the interpretations and conclusions justified by the results?

Yes

Is the language acceptable?

Yes

Do you have any ethical concerns with this paper?

No

Have you any concerns about statistical analyses in this paper?

No

Recommendation?

Accept with minor revision (please list in comments)

Comments to the Author(s)

This is very good piece of work, and suitable for publication, however; the reported yields for the conversion of the substrates were given without any standard deviation. In other words, the yield for olefins need to have standard deviations.

Decision letter (RSOS-191165.R0)

02-Sep-2019

Dear Professor Kilyanek:

Title: Deoxydehydration of vicinal diols by homogeneous catalysts: A mechanistic overview
Manuscript ID: RSOS-191165

Thank you for submitting the above manuscript to Royal Society Open Science. On behalf of the Editors and the Royal Society of Chemistry, I am pleased to inform you that your manuscript will be accepted for publication in Royal Society Open Science subject to minor revision in accordance with the referee suggestions. Please find the reviewers' comments at the end of this email.

The reviewers and handling editors have recommended publication, but also suggest some minor revisions to your manuscript. Therefore, I invite you to respond to the comments and revise your manuscript.

Because the schedule for publication is very tight, it is a condition of publication that you submit the revised version of your manuscript before 11-Sep-2019. Please note that the revision deadline will expire at 00.00am on this date. If you do not think you will be able to meet this date please let me know immediately.

Supplementary files will be published alongside the paper on the journal website and posted on

the online figshare repository (<https://figshare.com>). The heading and legend provided for each supplementary file during the submission process will be used to create the figshare page, so please ensure these are accurate and informative so that your files can be found in searches. Files on figshare will be made available approximately one week before the accompanying article so that the supplementary material can be attributed a unique DOI.

Best wishes,
Dr Laura Smith
Publishing Editor, Journals

RSC Associate Editor:
Comments to the Author:
(There are no comments.)

RSC Subject Editor:
Comments to the Author:
(There are no comments.)

Reviewer comments to Author:
Reviewer: 1

Comments to the Author(s)

Kilyanek and coworkers present in their manuscript "deoxydehydration of vicinal diols by homogeneous catalysis: A mechanistic overview" a relatively narrow chapter of deoxydehydration of polyols to generate alkenes and dienes chemistry, but in a very comprehensive. The review is quite well written and the collection of literature is very well presented. The authors described the mechanistic pathways for these DODH reactions. The description and representation of the examples in the manuscript does not need to be criticized. It is rather the selection.

Given the topical interest, with the practical importance of sustainable deoxydehydration reactions, I recommend publication of this fine manuscript after the following revision has been corrected.

1) Scheme 9, the drawing could be corrected.

2) Reference 42 and 61 are same: this need to be corrected.
There are several minor formatting errors throughout the manuscript which could be corrected by careful editing.

Reviewer: 2

Comments to the Author(s)

This is very good piece of work, and suitable for publication, however; the reported yields for the conversion of the substrates were given without any standard deviation. In other words, the yield for olefins need to have standard deviations.

Author's Response to Decision Letter for (RSOS-191165.R0)

See Appendix A.

Decision letter (RSOS-191165.R1)

04-Oct-2019

Dear Professor Kilyanek:

Title: Deoxydehydration of vicinal diols by homogeneous catalysts: A mechanistic overview
Manuscript ID: RSOS-191165.R1

It is a pleasure to accept your manuscript in its current form for publication in Royal Society Open Science. The chemistry content of Royal Society Open Science is published in collaboration with the Royal Society of Chemistry.

RSC Associate Editor
Comments to the Author:
The manuscript can now be accepted.

Reviewer(s)' Comments to Author:

Appendix A

Dear Dr. Smith,

Enclosed is our revised manuscript "Deoxydehydration of vicinal diols by homogeneous catalysts: A mechanistic overview". We have addressed the comments of the reviewers and our point by point response is given below.

Regards

Stefan M. Kilyanek

Reviewer comments to Author:

Reviewer: 1

Comments to the Author(s)

Kilyanek and coworkers present in their manuscript "deoxydehydration of vicinal diols by homogeneous catalysis: A mechanistic overview" a relatively narrow chapter of deoxydehydration of polyols to generate alkenes and dienes chemistry, but in a very comprehensive. The review is quite well written and the collection of literature is very well presented. The authors described the mechanistic pathways for these DODH reactions. The description and representation of the examples in the manuscript does not need to be criticized. It is rather the selection.

Given the topical interest, with the practical importance of sustainable deoxydehydration reactions, I recommend publication of this fine manuscript after the following revision has been corrected.

1) Scheme 9, the drawing could be corrected.

Scheme 9 was corrected to include water produced upon reduction and labeling the Re(V) intermediate species as Re(V)(O)(diolate).

2) Reference 42 and 61 are same: this need to be corrected.

There are several minor formatting errors throughout the manuscript which could be corrected by careful editing.

reference 61 is removed and the references renumbered. The manuscript has undergone another round of proofreading.

Reviewer: 2

Comments to the Author(s)

This is very good piece of work, and suitable for publication, however; the reported yields for the conversion of the substrates were given without any standard deviation. In other words, the yield for olefins need to have standard deviations.

We agree with reviewer 2 that standard deviations for the conversions and yields would be ideal. In fact, in our own work we attempt to report these when possible. After a thorough search of all of the references and their SI (where available), no standard deviations are provided. In a few examples, yields and conversions are reported as averages of several runs. However, no standard deviations, data or tables of the raw reaction info has been provided. Additionally, none of the other reviews cited here contain this information.